# Isolation and Phenotypic Microarray Profiling of Different *Pseudomonas* Strains Isolated from the Rhizosphere of *Curcuma longa* L.

**Parul Pathak** [1,†]**, Monika Singh** [2,†]**, Ananya Naskar** [2]**, Sandeep Kumar Singh** [3]**, Nikunj Bhardwaj** [4] **and Ajay Kumar** [5,*]

1   Department of Microbiology, Singhania University, Pacheri Bari, Jhunjhunu 333515, India; parulpathak2270@gmail.com
2   Department of Biotechnology, School of Applied and Life Sciences, Uttaranchal University, Dehrdaun 248007, India; monikasingh_bhu@yahoo.com (M.S.); naskar.ananya2003@gmail.com (A.N.)
3   Division of Microbiology, ICAR-Indian Agricultural Research Institute, Pusa, New Delhi 110012, India; sandeepksingh015@gmail.com
4   Department of Zoology, Maharaj Singh College, Maa Shakumbhari University, Saharanpur 247001, India; bhardwajnikunaj8@gmail.com
5   Amity Institute of Biotechnology, Amity University, Noida 201313, India
*   Correspondence: ajaykumar_bhu@yahoo.com; Tel.: +91-8960639724
†   These authors contributed equally to this work.

**Abstract:** In the present study, different *Pseudomonas* strains were isolated from the rhizospheric soil of *Curcuma longa* (turmeric) and further identified and characterized based on morphological, biochemical, and molecular characteristics through the 16S rRNA gene sequencing analysis. The identified bacterial strains belong to the *Pseudomonas* genus viz. *Pseudomonas* sp. CL10, *Pseudomonas* sp. CL11, and *P. fluorescence* CLI4. However, the isolated strains tested positive for IAA production, siderophore production, and the solubilization of tricalcium phosphate during plant growth promoting traits analysis. Further phenotype microArray (PM) technology was used to evaluate the antibiotic and chemical sensitivity of the isolated bacterial strains. The antibiotics phleomycin, oxacillin, vancomycin, novobiocin, spiramycin, and rifampicin, as well as chemicals like, 5-7 dichloro-8-hydroxy quanaldine, 5-7 dichloro-8-hydroxyquinoline, domophenbrobide, and 3-5 dimethoxy benzyl alcohol, showed resistance in all the rhizobacterial strains. However, upon further detailed study, *Pseudomonas* sp. CL10 exhibited resistance to thirteen antibiotics, CL11 to fourteen, and CL14 showed resistance against seventeen antibiotics and chemical classes. The results of the study indicate that some of these strains can be used as bioinoculum to enhance the plant growth and health.

**Keywords:** turmeric; rhizosphere; plant growth promotion; 16S rRNA; antibiotic resistance; Biolog; PM plate





## 1. Introduction

Food security is one of the major concerns for the rising global population. However, to meet the requirement of food and to enhance agricultural productivity, a large population relies on chemical fertilizers or chemical pesticides. But the continuous and undistributed use of these agrochemicals adversely affects the texture and productivity of soil, nutrient quality of fruits, and the native soil microflora [1]. Therefore, there is an urgent need to find alternative strategies that can ensure competitive crop yields and provide environmental safety and protection while maintain long term ecological balance in agro-ecosystem.

Plant growth-promoting rhizobacteria (PGPR) are the natural inhabitant's bacterial species that effectively colonize the roots or rhizosphere of the plant and modulate their growth and productivity, even under the stress conditions. These bacterial species play a

crucial role in phytohorome modulation, nutrient acquisition, and phytopathogen management that are an essential requirement for the plant growth. In last few years PGPR strains have been frequently used as bio stimulants, inoculants, or microbial fertilizers to enhance the agricultural productivity [2–4].

Plants secrete significant amount of root exudates, which is constituted of different amounts of amino acids, carbohydrates, lipids, flavonoids, and other biochemical compounds. These exudates act as chemo attractants or signaling molecules for the microbes and facilitate effective root colonization [5–7]. However, compositions of the root exudates are specific for each plant species and the response of the microbial strains towards the exudates are also specific [8,9]. The rhizosphere of the plant generally refer to a thin zone around the roots that adheres with the soil [10,11]. The rhizosphere is considered as a dynamic region of plant-microbe interaction, where various types of positive or negative interactions take place between the plant, soil, and microbes [12–14].

*Curcuma longa* L., commonly known as turmeric, is a rhizomatous herb and one of the most prominent species of the Zingiberaceae family, frequently used as a spice in the daily cuisine of the Indian subcontinent [15]. In addition, since ancient times, turmeric has also been used as a traditional medicine in south Asian countries, due to its pharmaceutical importance, especially curcumin or sesquiterpenoids [15,16]. For an example, curcumin one of the important constituents of turmeric have been extensively used to treat various human ailments such as bronchial asthma, jaundice, abdominal cramps, chronic hepatic diseases, melanoma, urinary tract infections, and many more [17–19].

The rhizosphere of the turmeric also interacts with diverse microbial communities; in previous studies various authors reported different epiphytic and endophytic bacterial strains and their significance in plant growth promotion and phytopathogen management [17–19]. *Bacillus, Pseudomonas, Azotobacter,* etc., are the common rhizospheric strains of the turmeric rhizosphere, which has shown excellent plant growth promotion potential as reported in the previous studies [19,20]. In case of turmeric, the *Pseudomonas fluorescens* bacterial strains isolated from the rhizospheric soil of turmeric enhance the plant growth and curcumin content significantly after inoculation [20]. As well as *Pseudomonas* strains, various microbial strains of genera *Bacillus, Pseudomonas, Trichoderma, Pantoea,* etc., have been frequently used as biofertilizers, as reported in previous studies [21–24]. However, in another study, Shaharoona et al. [25] observed significant enhancement in the root weight, and straw and grain yield of wheat after inoculation of *Pseudomonas fluorescens*. Similarly, Arshad et al. [26] reported inoculation of *Pseudomonas* spp. significantly enhanced the growth and yields of pea plant.

However, for a PGPR-based inoculant to be effective, it is essential that inoculants or the bacterial strains exhibit not only plant growth-promoting characteristics but also the ability for economical mass production and long-term stability in formulation [4]. While some PGPR strains have been found to harbor antibiotic resistance genes (ARGs), the extent of the associated risks and the potential transmission of ARGs to soil microorganisms, as well as the impact on bacteria associated with humans and the food chain, remain largely unexplored [27,28]. Nowadays, a phenotype microarrays (PM) technique has been used to determine the antibiotic/chemical sensitivities of any microbial strains via comparing phenotyping gene functions or to studying metabolic properties. The phenotypic response of the microbial cells can be visualized by observing changes in color and turbidity of the PM plates [29].

Therefore, considering nutritional and medicinal importance of the turmeric plant, the objective of this investigation is to segregate diverse strains of *Pseudomonas* bacteria and evaluate their capacity to promote plant growth, as well as their potential to tolerate salinity. Furthermore, we will conduct assays to determine the long-term suitability of these isolates for practical applications in the field, specifically by employing PM plates to assess their sensitivity to antibiotics and chemicals.

## 2. Result and Discussion

On the basis of morphological dissimilarity total of nine bacterial isolates were selected from the bacterial growing plates. Further on the basis of morphology, biochemical characterization and excellent growth on the King's B agar medium, three isolates, CL10, CL11, and CL14, were considered for further studies (Figure 1).

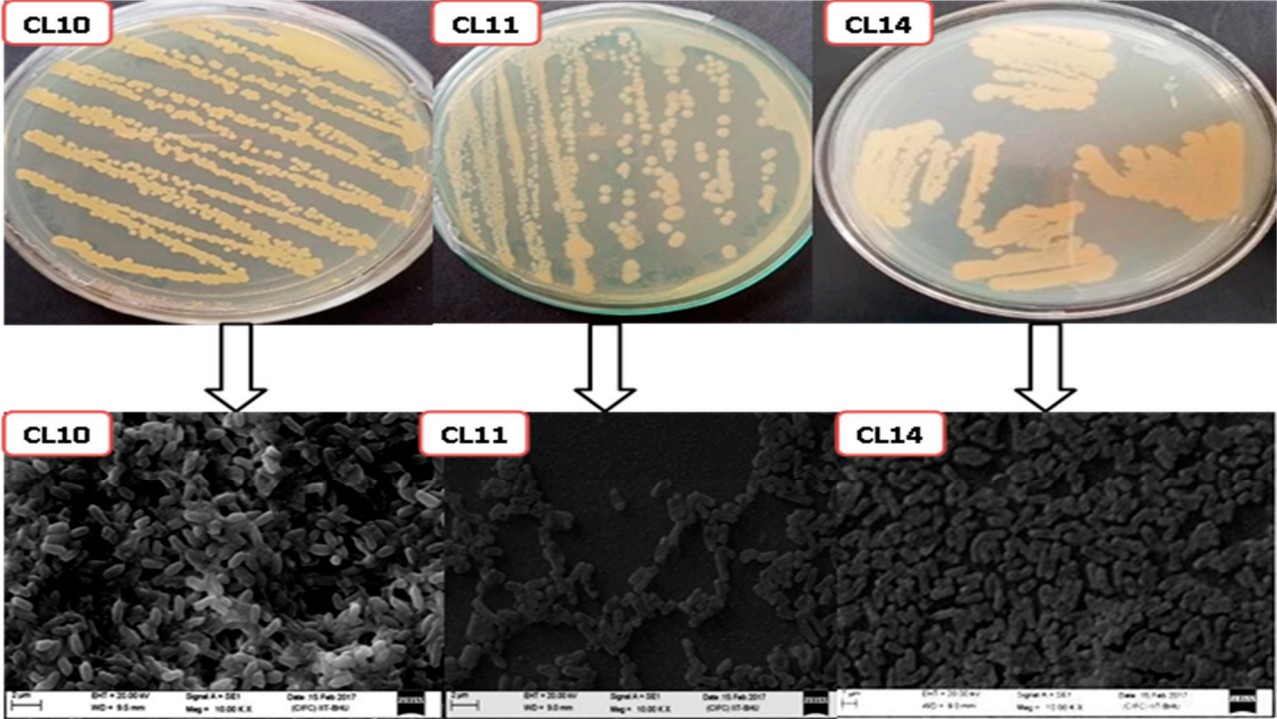

**Figure 1.** Colony morphology of isolated bacterial strains on agar plates and scanning electron micrographs of *Pseudomonas* sp., strains CL10, CL11, and CL14. The scale bar is 1 μm.

All three isolates were Gram-negative and rod-shaped in microscopic observation. During biochemical analysis, all the strains were positive for catalase, oxidase, glucose, and starch hydrolysis tests details of the analysis were summarized in Table 1.

**Table 1.** Biochemical characterization of rhizospheric *Pseudomonas* bacterial isolates of turmeric.

| Strain | Gram Staining | Shape | Catalase | Oxidase | Glucose | Lactose | Maltose | Mannitol | D-Mannose | Sucrose | Nitrate Reduction | H2S Production | Starch Hydrolysis |
|---|---|---|---|---|---|---|---|---|---|---|---|---|---|
| CL10 | − | Rod | + | − | + | − | + | + | − | + | − | − | + |
| C L11 | − | Rod | + | + | + | − | − | + | − | + | − | − | + |
| CL14 | − | Rod | + | + | + | − | − | − | + | − | − | − | + |

(+: Positive, −: Negative).

However, after 16S rRNA gene sequence analysis, these three isolates were assigned as *Pseudomonas* sp. CL10, *Pseudomonas* sp. CL11, and *Pseudomonas fluorescence* CL14.

Their percentage similarity of the 16S rRNA and the GeneBank accession number have been mentioned in Table 2 and phylogenetic analysis using neighbor-joining method were displayed in Figure 2.

**Table 2.** Nearest relatives of the identified rhizospheric bacterial strains data obtained by 16S rRNA gene sequencing analysis.

| Identified Isolate | Accession Number | Nearest Phylogenetic Neighbor Strain | Percent of Similarity |
|---|---|---|---|
| CL10 | KM067137 | *Pseudomonas* sp. strain BHUJPV-A13 | 100% |
| CL11 | KM067136 | *Pseudomonas* sp. strain DKF | 100% |
| CL14 | KM067138 | *Pseudomonas fluorescens* strain ESR7 | 100% |

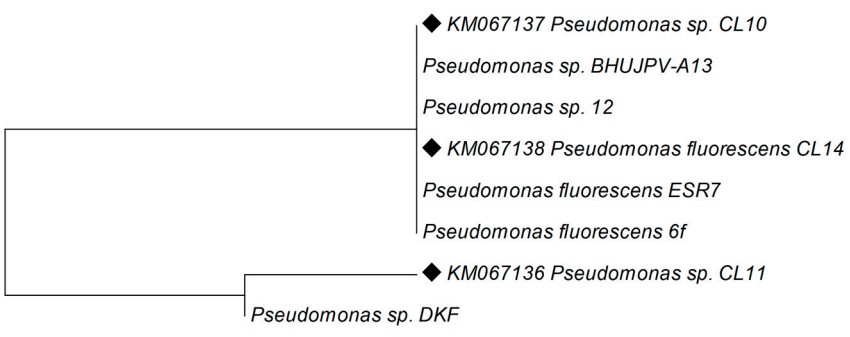

**Figure 2.** Phylogenetic tree from analysis of 16S rRNA gene sequence of the rhizospheric strains of *C. longa* L. GenBank accession numbers of nucleotide sequences are shown along with the name of bacterial strain. Phylogenetic analyses were conducted in MEGA 4.1.

## 2.1. Plant Growth Promotion (PGP) Activity

During PGP traits analysis, all the strains significantly synthesized IAA, solubilized phosphate, and produced siderophore and ammonia. The strain *Pseudomonas* C10 was a strong producer of IAA, (Table 3). However, all the three isolates show differential level of salt tolerance. *Pseudomonas* sp.CL11 showed tolerance up to 5% whereas *Pseudomonas fluorescence* CL14 showed 6% tolerance of NaCl.

**Table 3.** PGP traits analysis of bacterial isolates.

| Identified Isolate | Indole-Acetic Acid Production | Phosphate Solubilization | Ammonia (NH3) Production | Siderophore Production | Salinity Tolerance |
|---|---|---|---|---|---|
| *Pseudomonas* sp. CL10 | + | + | + | + | 5% |
| *Pseudomonas* sp. CL11 | + | + | + | + | 5% |
| *Pseudomonas fluorescence* CL12 | + | + | + | + | 6% |

## 2.2. Antibiotic and Chemical Sensitivity Using Phenotypic Microarray Technology

Phenotypic MicroArray (PM) technology (Biolog, Springfield, MO, USA) provides a unique and easy methods to identify antibiotic resistance. In these assays, the chemicals are pre-filled and dried in 96-well microplates that can monitor in terms of sensitivities [30,31]. Recently, various studies used PM technology to explore the gene functions, and biochemical and molecular characteristics of microbes [30–32]. In the present study, the PM 12B and PM 15B plates were used to evaluate the antibiotic and chemical sensitivity of *Pseudomonas* strains CL10, CL11, and CL14.

Antibiotic and chemical sensitivity was observed using PM (Phenotypic Microarray) 12B and PM15C plates of Biolog Microstation system. The heat maps were also constructed on the basis of their mean value of absorbance (Figure 3).

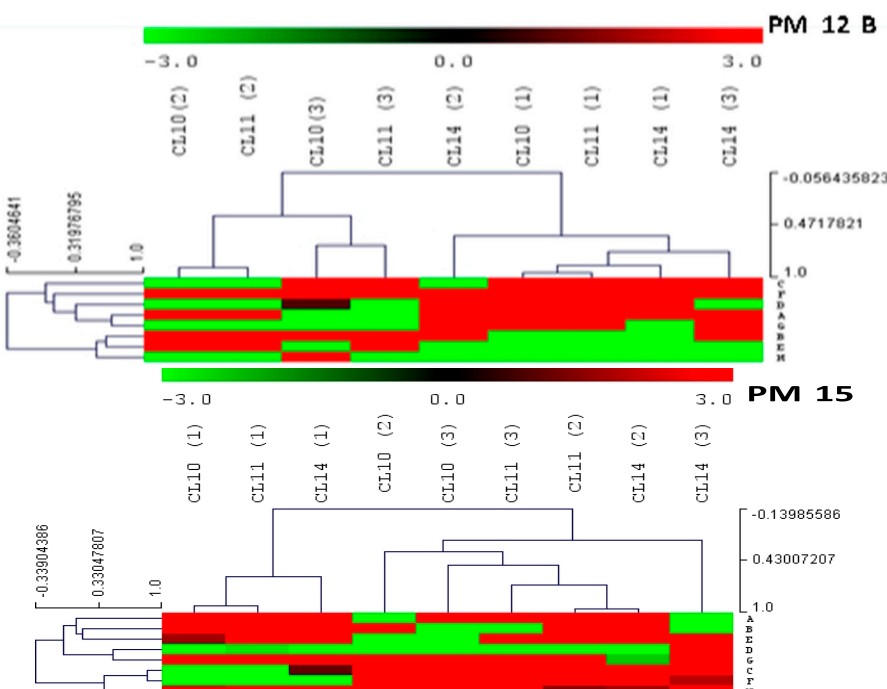

**Figure 3.** Heat map dendrogram on the basis of antibiotic and chemical sensitivity of rhizobacterial isolates CL10, CL11, and CL 14 after 48 h of incubation (Mean value of four wells was taken to construct the heat map).

The heatmap was constructed with mean values of absorbance taken from the substrates present the well of PM 12B plate. The dendrogram of the heat map is divided into two main clusters, and both clusters are further divided into two sub-clusters (Figure 3). Cluster one formed two sub-clusters with substrates utilized by CL11 and CL10, while Cl14 was included in cluster two with CL10 and CL11. The substrates penicillin G, tetracycline, penimepicycline, polymyxin B, paromomycin, D-L serine hydroxamate, sisomicin, tobramycin, sulphathiazole, 5-flouroorotic acid, and spectinomycin showed strong color intensity by all three isolated strains, while di isopropyl pteridine sulphadiazine showed strong effect by CL10 and CL11, but not with CL14. The carbenicillin, sulfamethazine, sulphamethoxazole, and 1-aspartic acid b-hydroximate showed strong color intensity by only *P. fluorescence* CL14 [33]. PM technology has been also used to identify the differences between microbial species [34]. Only substrates dodecyl trimethyl ammonium bromide and benzethonium chloride showed positive results by CL10 and CL11, respectively. Detailed results have been presented in Table 4 and Figure 4.

In the study, PM12 plates were also used for antibiotic sensitivity analysis of eight major environmental strains such as *Acinetobacter* sp., and *Microbacterium* sp. was also evaluated [35]. *Streptococcus thermophiles* strain was also evaluated using PM technology for their chemical sensitivity [36].

The PM15B test panel of antibiotics and chemicals was also used and discussed with the help of heatmap dendrogram. This dendrogram is also divided into two sub-clusters and these were further divided into two sub-clusters. The heat map clusters indicated that CL10 and CL11 are more closely related species compare with *P. florescence* CL14. The cluster one divided into two sub-clusters: sub-cluster one construct with substrate utilized by CL10 and CL11, while sub-cluster two of cluster one formed with substrates utilized by only CL 14. The second large cluster also divided into two sub-clusters. Sub-cluster one also included the more substrates utilized by CL10 and CL 11 and few substrates utilized by CL 14 and sub-cluster two of cluster two made with CL 14 alone. The chemicals procaine, D–cycloserine, EDTA, fusidic acid, 1-10 phenanthroline, oleandomycin, carbonyl cyanide m-chloro phenyl hydrazine (CCCP), menadione, 2-nitroimidazole, and zinc chloride showed strong intensity with all three rhizospheric strains of turmeric. Mean-

while, cefmetazole, sodium azide, puromycin, showed positive results with CL10 and CL11, and guanidium hydrocholoride, alexidine, 5- nitro-2- furaldehydesemicarbazone, and methyl viologen were sensitive with CL11 and CL14. The two substrates hydroxyl urea and nordihydroguaia acid showed positive with CL10 and CL14, respectively, whereas 5-7 dicholoro-8 hydroxyqunoline, phleomycin, domiphen bromide, and 3-4 dimethoxy benzyl alcohol showed less or no color intensity with the all rhizospheric isolates. *Pseudomonas cichorii* isolated from turmeric (*C. longa* L.) was analyzed and identified with similarity index of 84.2% and a probability of 100% in country Brazil [37].

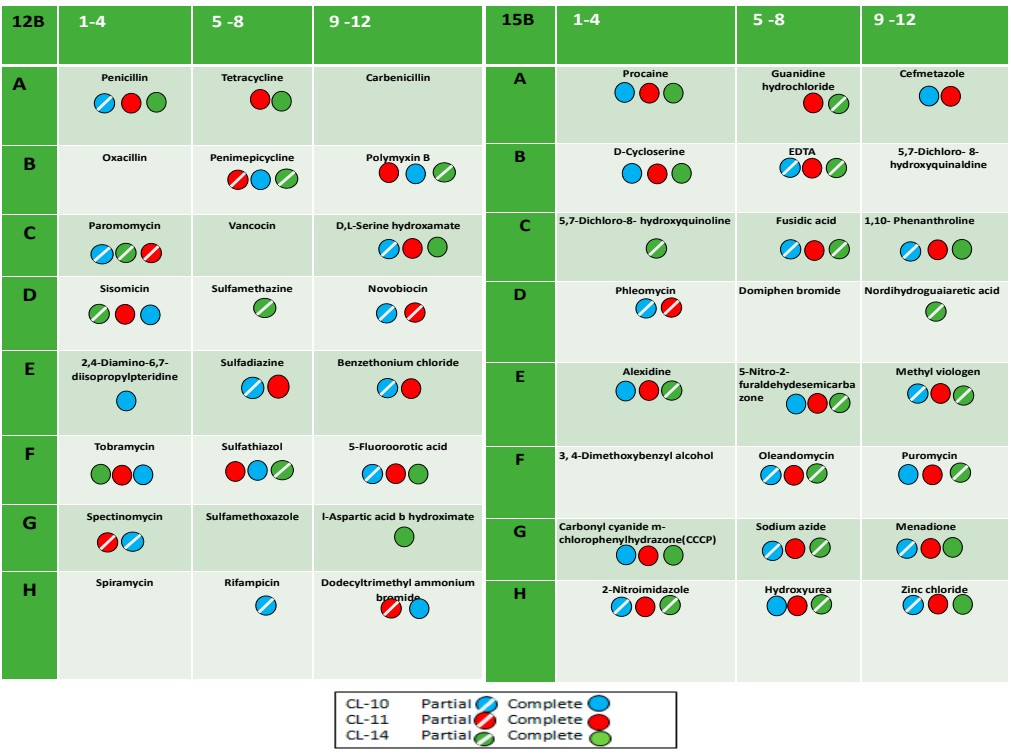

**Figure 4.** Antibiotic and chemical assay of rhizospheric bacterial strains: The figure showed complete and partial utilization of antibiotics and chemical on 12B and 15B PM plates of all the three rhizobacterial strains.

Utilization of plant growth promoting rhizobacteria (PGPR) is one of the emerging approaches to enhanced plant growth or phytopathogen management in sustainable agriculture. Recently, various PGPR strains have been isolated and characterized or used to enhanced agricultural productivity [6,14]. However, various species of *Pseudomonas* have been also considered as suitable bio fertilizer due to high colonization efficacy and secretion of various antimicrobial compounds that directly or indirectly associated with plant growth promotion and phytopathogen control as reported in previous studies [4,38]. In our study, all the three *Pseudomonas* strains showed PGP properties including siderophore production and phosphate solubilization. In previous studies, authors also reported PGP attributes in different strains such as *P. fluorescens*, *P. putida*, and *P. syringae* [39].

Phosphate solubilization is one of the crucial traits of plant growth promotion through which bacterial isolates solubilized the insoluble tri-calcium phosphate to soluble and make them available to plants. In the natural state, phosphorous is in monobasic ($H_2PO^-$) and dibasic ($HPO^-$) soluble form which is utilizable by the plants. In addition, phosphate solubilizing bacteria secrete low molecular weight organic acids like oxalic-, Citric-, and Malic acid which are also involved in modulating plant growth [40].

**Table 4.** Antibiotic and chemical assay using PM 12B and 15B microplates with their mode of actions.

| Well No. in PM 12 B Plate | Compounds and Their Mode of Action | | Well No. in PM 15B Plate | Compounds and Their Mode of Action | |
|---|---|---|---|---|---|
| A1 (1–4) | Penicillin | Inhibiting cell wall | A1 (1–4) | Procaine | Inhibiting sodium influx through voltage gated sodium channels |
| A2 (5–8) | Tetracycline | Inhibit protein synthesis, prevent the association of aminoacyl-tRNA with ribosomes | A2 (5–8) | Guanidine hydrochloride | Chaotropic agent |
| A3 (9–12) | Carbenicillin | Inhibit final cell wall synthesis of susceptible bacteria. | A3 (9–12) | Cefmetazole | Inhibition of cell wall synthesis |
| B1 (1–4) | Oxacillin | Block the peptidoglycan synthesis | B1 (1–4) | D-Cycloserine | Inhibiting cell-wall biosynthesis |
| B2 (5–8) | Penimepicycline | A bactericidal action on the streptococci, and a bacteriostatic action on various gram-positive and gram-negative bacilli | B2 (5–8) | EDTA | Chelating agent |
| B3 (9–12) | Polymyxin B | Interact with lipopolysaccharide (LPS) of the outer membrane of Gram -ve bacteria | B3 (9–12) | 5,7-Dichloro-8-hydroxyquinaldine | Induces SOS-DNA repair |
| C1 (1–4) | Paromomycin | Inhibit protein synthesis via ribosomal subunits interaction | C1 (1–4) | 5,7-Dichloro-8-hydroxyquinoline | Inhibits the growth of Gram-positive and some Gram-negative bacteria |
| C2 (5–8) | Vancocin | Wall, lactam | C2 (5–8) | Fusidic acid | Interfering with bacterial protein synthesis, |
| C3 (9–12) | D,LSerin hydroxamate | tRNA synthetase | C3 (9–12) | 1,10-Phenanthroline | Inhibition of carboxypeptidase |
| D1 (1–4) | Sisomicin | Binding to small ribosomal subunit, inhibit the protein synthesis | D1 (1–4) | Phleomycin | DNA breakage |
| D2 (5–8) | Sulfamethazine | Folate antagonist, PABA analog | D2 (5–8) | Domiphen bromide | Cationic surfactant |
| D3 (9–12) | Novobiocin | DNA topoisomerase | D3 (9–12) | Nordihydroguaiaretic acid | Inhibit arachidonic acid 5-lipoxygenase activity |
| E1 (1–4) | 2,4-Diamino-6,7-diisopropylpteridine | Antifolate (disruption of the metabolic pathways) | E1 (1–4) | Alexidine | Binds to lipopolysaccharide and lipoteichoic acid |
| E2 (5–8) | Sulfadiazine | Antifolate | E2 (5–8) | 5-Nitro-2- furaldehydesemicarbazone | Inhibits several bacterial enzymes |
| E3 (9–12) | Benzethonium chloride | Detergent | E3 (9–12) | Methyl viologen | Electron relay in photochemical systems |
| F1 (1–4) | Tobramycin | Protein synthesis, 30S ribosomal subunit, aminoglycoside | F1 (1–4) | 3, 4-Dimethoxybenzyl alcohol | Used as the fuel of the *microbial fuel cell* |
| F2 (5–8) | Sulfathiazole | Antifolate, PABA analog | F2 (5–8) | Oleandomycin | Binds to the 50s subunit of bacterial ribosomes |
| F3 (9–12) | 5-Fluoroorotic acid | Nucleic acid analog, pyrimidine | F3 (9–12) | Puromycin | Analogue of the 3′ end of aminoacyl-transfer RNA |
| G1 (1–4) | Spectinomycin | Inhibit protein synthesis with the binding of 30S ribosomal subunit, | G1 (1–4) | Carbonyl cyanide m-chlorophenylhydrazone(CCCP) | Inhibitor of oxidative phosphorylation |
| G2 (5–8) | Sulfamethoxazole | Antifolate, PABA analog | G2 (5–8) | Sodium azide | Inhibits activities of peroxidases, catalase and few more enzymes |
| G3 (9–12) | l-Aspartic acid b-hydroxamate | tRNAsynthetase | G3 (9–12) | Menadione | As a cofactor in the posttranslational gamma-carboxylation |
| H1 (1–4) | Spiramycin | Inhibits translocation by binding to bacterial 50S ribosomal subunits | H1 (1–4) | 2-Nitroimidazole | Inhibits protein synthesis and RNA synthesis |
| H2 (5–8) | Rifampicin | Inhibit bacterial DNA-dependent RNA polymerase | H2 (5–8) | Hydroxyurea | Inhibition of the enzyme ribonucleotide reductase |
| H3 (9–12) | Dodecyl trimethyl ammonium bromide | Surfactant | H3 (9–12) | Zinc chloride | Catalyst in chemical metal |

Modulation of phytohormone through use of PGPR is one of the important aspects of plant growth promotion. Various phytohormones such as IAA, cytokinin, and gibberellins have been synthesized by PGPR strains [4]. In our study, all three strains synthesize Indole acetic acid. Usually, strains producing IAA stimulate plant growth, seed germination, and protect the plants from biotic and abiotic stresses [41]. In addition, the IAA synthesizing bacteria colonize the plant interiors and stimulate plant growth [42].

The siderophore, an iron chelating compound present in the soil that binds with ferric ions (Fe+3), is taken up by microbial cell through specific recognition bmembrane proteins [43]. The presenceof siderophores makes the microbes better competitors for iron acquisition and in this way prevents growth of the pathogenic microbes. In a previous study, Laslo et al. [44] reported that phosphate solubilization, IAA production, and siderophore production were reported in *Pseudomonas* sp. [44,45]. The *Bacillus* and *Pseudomonas* sp., isolated from *Curcuma longa* L., solubilize phosphate, and produce IAA and siderophore during PGP trait analysis. Both the strains also utilized various sources of carbon such as glucose, sucrose, and yeast extract, whereas glycine, alanine, cystine, and glutamine have been utilized as nitrogen source [12].

In the current survey, approximately more than 20% of agricultural land faces challenges of salinity stress. The presence of saline condition severely affects the growth, or even survivability, of plants due to poor root growth [46]. It is assumed that alkalinity (pH < 8.5) present in the soil severally affects the plant [47]. In the present study, *Pseudomonas* sp. CL10 and CL11 tolerated maximum salinity tolerance up to the 5%, with *Pseudomonas fluorescence* CL14 demonstrating a tolerance of 6%. Rashid et al., [48] reported *P. fluorescence* strain which tolerated 4% NaCl. Similarly in a study strain *Pseudomonas* sp. M30-35 might confer salt stress tolerance to *Chenopodium quina* wild during growth conditions [49].

Phenotypic Microarray (PM) technology is an advanced technique of microbial growth [50]. This PM technology has high enough sensitivity that it enables various tests to be performed even if the samples or microbial cultures are only present in small quantities. Through the PM assays, various analysis such as antibiotics and specific chemicals can be performed in a very compact space in much less time than conventional methods. In the previous studies, through PM assay, chemical sensitivity of the microbes has been already performed [51–53]. A study by Santopolo et al. [54] used the PM technology along with Calgary Biofilm device for the characterization and sensitivity of *Pseudomonas alcaliphila* strain. Through this study, it has been observed that all three isolates showed plant growth promoting traits, but strains CL14 showed resistance against the different antibiotics and chemicals. This strain, after long-term utilization as a bio fertilizer, may cause transfer of antibiotic resistance gene to the native inhabitant microbial communities.

The primary obstacles in utilizing PGPR as biofertilizers is ensuring their stability and enhancing their survivability to establish a competitive presence in the soil with existing microbial community [55]. Additionally, PGPR strains with inherent antibiotic resistance can be employed as indicators to evaluate the viability of introduced bacteria after application [56]. Hence, there is an urgent need to isolate and screen the microbial strains that possess specific traits to ensure their successful adaptation to various environments [57]. Previous study revealed successful implementation of different plant growth promoting rhizobacterial strains in plant growth promotion, phytopathogen management, or different abiotic or biotic stress management. However, the PGPR strains having potential of multi antibiotic resistance do not adequately address the biosafety concerns associated with the potential spread of multi antibiotic resistance gene to the soil, native microbial communities, or the food chain after used as soil or plant inoculants [1]. It has been reported on the basis of sequencing data that most of the plant growth promoting bacterial strains has the characteristic of showing resistance against one of the antibiotics or some genes of antibiotic resistance [58]. In a previous study, it has been reported that inoculation or treatment of antibiotic-resistant bacteria as biostimulants or biofertilizers can be transferred

their antibiotic resistance genes to the native microbial species, soil, or the water which ultimately affect the food chains [28].

## 3. Materials and Methods

### 3.1. Isolation of Rhizospheric Bacteria

For the isolation of bacterial isolates, we collected rhizospheric soil from turmeric plants (*C. longa* L.), which is growing on the Banaras Hindu University campus in India. In detail, 1 g of rhizospheric soil was collected with a sterile brush and serially diluted to create dilution ranging from $10^{-5}-10^{-8}$. Afterward, 0.1 mL of the soil suspension was spread on the King's B media plates and incubated at 30 °C for 48–72 h. We had used only King's B media, because previous studies have shown that *Pseudomonas* strains exhibit the best plant growth promotion [20]. Subsequently, colonies that appeared on the plates were randomly selected and subculture to obtain pure isolates, which were then maintained on agar slants at 4 °C for further study.

### 3.2. Characterization of the Bacterial Isolates

The morphology and biochemical characterization of the bacterial isolates was performed according to standard protocols or the standard Bergey's Manual of Determinative Bacteriology [21,59]. In addition, scanning electron microscopy was also performed to visualize the colony morphology [60].

### 3.3. Scanning Electron Microscopy Analysis of Bacterial Strains

Samples for SEM were prepared using the standard protocol for biological samples. The bacterial samples were subjected to primary fixation in 2.5% glutaraldehyde with phosphate buffer. The bacterial sample slides were washed with 0.1% M buffer. After that, bacterial slides were dehydrated at 4 °C through the following steps: 30%, 50%, 70%, 80%, 90%, and 100% ethyl alcohol for 15 min each. Samples from 100% ethyl alcohol were transferred into a 1:2 solution of ethyl alcohol and Hexamethyldisilizane (HMDS) for 15 min. They were then transferred into 100% HMDS for 20 min, repeating until the samples were submerged in the solution, and then left overnight under the fume hood with loose cap. All these steps involving HMDS were carried out under the fume hood (ZEISS, Oberkochen, Germany, EVO 18) [61].

### 3.4. Plant Growth Promoting (PGP) Traits Analysis of Bacterial Isolates

*Production of IAA*: To estimate IAA production, bacterial broth cultures grown in LB media were supplemented with different concentrations of L-tryptophan (0 to 500 µg/mL). Afterward, cultures were centrifuged at 8000 rpm for 10 min. Then, 2 mL of bacterial supernatant was mixed with 2 drops of orthophosphoric acid ($H_3PO_4$) and 4 mL of the Salkowski reagent and incubated in the dark for 60 min. The development of a pink color confirmed the production of Indole-acetic-acid [62].

*Phosphate solubilization*: The isolated strains were spotted on the Pikovskaya's agar media plates and incubated at 30 °C for 48–72 h. The development of a clear halo zone around the bacterial spot confirmed the phosphate solubilization property [44].

Ammonia Production: The production of ammonia ($NH_3$) was assessed following the standard protocol of Marques et al. [63]. In brief, 20 µL of 24 h old culture was inoculated into 5 mL of 1% protease peptone broth and incubated at 30 °C in a shaking water bath. After 48 h, 0.5 mL of Nessler's reagent was added to the culture. The development of a brown to yellow color confirmed ammonia formation.

*Siderophore Production*: For the test, 48-h-old bacterial suspension was taken, and 5 µL was inoculated in the Chromeazurol S agar plate and incubated at 28 °C for 72 h. The development of a yellow to orange hollow zone around the bacterial spot was considered positive for siderophore production [64].

### 3.5. Salt Tolerance

For salinity stress tolerance, 20 μL aliquot of 24 h old bacterial culture was inoculated into Luria Bertini broth with different range of NaCl (0.5–10%), Further the salinity tolerance potential of the isolates was measured with UV-Vis spectrophotometer at 600 nm, after the interval of 24 h.

### 3.6. 16S rRNA Gene Amplification, Sequencing, and Their Analysis

The DNA of the bacterial isolates was extracted, amplified, and sequenced by following protocols and primers of Kumar et al. [20]. The taxonomic identification of the isolates was carried out by analyzing or comparing obtained 16S rRNA gene sequence with 16S rRNA partial sequence, available on the NCBI GenBank database with the BLAST search (www.ncbi.nlm.nih.gov/BLAST latest accessed on 25 June 2023) [20,30,64]. The comparative similarity with the bacterial gene sequence is above than 98% with the obtained gene sequence assigned as the bacterial taxonomy.

### 3.7. Evaluation of Antibiotic Sensitivity by Using PM Plates of Biolog Microstation System

Phenotypic microarrays (PM) Panels are 96-well microplates containing different antibiotics and chemical substrates in tetraplicate manner. In the present study, PM12B and PM15B microplates have been used to perform antibiotic and chemical sensitivity assay. In the 96-well plates, each well of the plate contains minimal medium components and a specific dye along with a unique substrate, whether it is antibiotic or specific chemical in a tetraplicate manner. In detail 24-h-old grown isolates on King's B agar medium were aseptically picked up and transferred into a sterile capped tube containing 20 mL of the inoculation fluid (IF-0a, Biolog Inc.). The bacterial density was maintained by adjusting 85% transmittance with the help of Biolog turbidimeter. Then the plates PM12B and PM15B were inoculated with the bacterial cell suspension and making the volume of each well up to 100 μL and incubated for 48 h in the Omnilog Incubator (Biolog Inc., Hayward, CA, USA) at the temperature of 30 °C. The changes in the color (white to pink and purple) of the wells were observed after every 12 h to visualize their growth pattern. All the Antibiotics and chemicals were present in tetraplicate manner. The mean value of absorbance was taken to construct the heatmap dendrogram by Meve software ver 16.0 [32].

### 3.8. Statistical Analysis

The molecular data or the gene sequence data of 16s rRNA sequence were aligned using Clustal X2.1 version and the phylogenetic analysis or neighbor-joining method using distance matrix were performed using the software MEGA 4.1. However, the heatmap analysis of the antibiotics and chemical sensitivity assay were prepared with Meve software package (ver. 16).

## 4. Conclusions

In the recent few years, the scenario of agricultural practices has been slightly changed, and now consumers are aware of the adverse consequences of chemical fertilizers and chemical pesticides, and, due to this reason, they demand organic food, the agricultural products which are the result of low use and no use of agrochemicals. In last two decades, plant growth promoting microorganisms including bacteria and fungi have been frequently used as plant or soil inoculants to enhance the agricultural productivity. However, survivability of these microbial inoculants after application is still a challenging task and only a small fraction of microbes survives and modulates the plant growth.

The present study describes isolation and characterization of plant growth promoting bacteria from the rhizospheric soil of *Curcuma longa* L. During the study three *Pseudomonas* strains have been isolated which showed excellent potential of plant growth promoting traits. In addition, the strains also showed tolerance against salinity stress. Further, the Phenotypic Microarray technology assay describes the sensitivity patterns of isolates against different antibiotics and chemicals. Through this study, the characterized strains of Pseu-

domonas can be used as plant or soil inoculants to enhance the agricultural production under normal or at the certain level of salinity stress. However, in this study, it has been observed that all three isolates showed plant growth promoting traits but strain CL14 showed resistance against the seventeen different antibiotics and chemicals. This strain, after long term utilization in bio fertilizers, may cause transfer of antibiotic resistance gene to the native inhabitant microbial communities, while the strain *Pseudomonas* sp. CL10 can be preferred in biofertlizers or bio stimulants in the sustainable agriculture practices.

**Author Contributions:** Conceptualization, M.S. and A.K. methodology, M.S. and A.K.; software, M.S. and S.K.S. validation, M.S., P.P., A.N., S.K.S., N.B. and A.K.; formal analysis, M.S. and A.K.; investigation, M.S.; resources, M.S.; data curation, M.S. and A.K.; writing—original draft preparation, M.S., P.P., A.N., S.K.S., N.B. and A.K.; writing—review and editing, M.S., A.N., S.K.S. and A.K.; visualization, A.K.; supervision, A.K.; project administration, M.S. and A.K.; funding acquisition, A.K. All authors have read and agreed to the published version of the manuscript.

**Funding:** This research received no external funding.

**Data Availability Statement:** Data are available within the article.

**Acknowledgments:** Authors are thankful to Amity Institute of Biotechnology, Amity university, Noida, India for providing lab facilities.

**Conflicts of Interest:** The authors declare no conflict of interest.

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
