# Peer review of "Isolation and Phenotypic Microarray Profiling of Different Pseudomonas Strains Isolated from the Rhizosphere of Curcuma longa L."

_stresses, doi:10.3390/stresses3040051_

Round 1

Reviewer 1 Report

Comments and Suggestions for Authors

I have reviewed the manuscript in which the authors studied "Isolation and Phenotypic Microarray Profiling of different Pseudomonas strains isolated from the rhizosphere of Curcuma longa L. "

I propose to the authors to be more specific, explanatory and simplified in order to be easily understandable from the readers. The authors need to clarify a few points throughout the manuscript. Here are some suggestions to back up my idea and help the author improve the quality of the paper:

Comment 1: Line. no. 23:  Change curcuma longa in to Curcuma longa.

Comment 2: Rewrite the abstract, it should be easy to understand the concept and conclusion.

Comment 3: Line no. 82, check the spelling of Pseudomonas fluorescents

Comment 4: There is a lot of research information available about the topic, but the introduction is not up to the level.

Comment 5: line no. 104, when you write the first time, full genus and species, when you subsequently do not need to mention the full name of the genus, just the first letter enough. Check throughout the manuscript.

Comment 5: lines no.105 and 106, and throughout the manuscript check the method of writing numerical values.

Comment 6: Avoid two lines as one paragraph.

Comment 7: Please rewrite the conclusion

Comment 8: Check English grammar and typo errors throughout the manuscript.

Author Response

Reviewers#1

I propose to the authors to be more specific, explanatory and simplified in order to be easily understandable from the readers. The authors need to clarify a few points throughout the manuscript. Here are some suggestions to back up my idea and help the author improve the quality of the paper:

Response: We are very thankful to the anonymous learned reviewer for their critical suggestions and comments. Definitely these suggestions improved the quality of our article. We have critically revised the article as per suggestions and the revised texts are highlighted in red.

Comment 1: Line. no. 23:  Change curcuma longa in to Curcuma longa.

Response: Done

Comment 2: Rewrite the abstract, it should be easy to understand the concept and conclusion.

Response: Done; we  have restructured the abstract.

Comment 3: Line no. 82, check the spelling of Pseudomonas fluorescents

Response: Done

Comment 4: There is a lot of research information available about the topic, but the introduction is not up to the level.

Response: We totally agree with reviewer remark; however as it is a small piece of work, so we have avoided to add advanced section. Definitely we will add latest developments in the next article.

Comment 5: line no. 104, when you write the first time, full genus and species, when you subsequently do not need to mention the full name of the genus, just the first letter enough. Check throughout the manuscript.

Response: Thanks for suggestion, we have revised it throughout the article

Comment 5: lines no.105 and 106, and throughout the manuscript check the method of writing numerical values.

Response: Thanks for suggestion, we  have revised it throughout the article

Comment 6: Avoid two lines as one paragraph.

Response: Thanks for suggestion, we have revised it

Comment 7: Please rewrite the conclusion

Response: Thanks for suggestion, we have revised and rewritten the conclusion section

Comment 8: Check English grammar and typo errors throughout the manuscript.

Response: Thanks for suggestion, we have thoroughly revised to avoid any errors

Reviewer 2 Report

Comments and Suggestions for Authors

The manuscript "Isolation and Phenotypic Microarray Profiling of different Pseudomonas strains isolated from the rhizosphere of Curcuma longa L." by Parul Pathak, Ananya Naskar, Monika Singh, Sandeep Kumar Singh, Manoj Kumar Solanki, Nikunj Bhardwaj and Ajay Kumar deals with the isolation and characterisation of strains of the genus Pseudomonas which exhibited properties characteristic of plant growth promoting rhizobacteria.

Comments:

Line 57-58. References 5,9 are inconsistent with the statement made. I recommend deleting this sentence completely as it has no scientific basis.

The introduction is poorly structured. For example, the definition of PGPR is on lines 46-48 and lines 80-82.

Line 88 is "PGPB" - correct the typo.

At the end of the introduction, there is no clearly stated purpose of the study.

Materials and methods.

P.2.6 - missing reference to sequences (e.g. GenBank) used for analysis.

Line 154 - reference 6 - reference numbering appears to be incorrect.

Figure 1 - photographs of colonies of microorganisms should not be shown. Furthermore, morphological characterisation of colonies is a very outdated method of characterising microorganisms.

Line 197 - 16 S rRNA is written in a slit - probably a typo.

Table 2. The given sequences are short and small enough to determine the species of the strains on the basis of them. It is correct to compare your sequences with the 16S rRNA gene sequences of the type strains.

Figure 2 does not show the scale.

Figure 3 is not informative, I recommend deleting it.

Figure 4 is of very poor quality and lacks transcripts A, B,... It is not clear what the authors are trying to show with these data.

It should be noted that the Results and Discussion section is also not very well structured. I recommend rewriting this section by breaking the text into parts with justification of why certain methods are used.

Overall conclusion: I recommend a thorough revision of the paper. Authors should clearly state the purpose of the research. It is also important that the paper lacks scientific novelty. I recommend that the authors decide on the novelty and purpose and rewrite the text of the paper accordingly.

Comments on the Quality of English Language

Moderate editing of English language required

Author Response

Reviewer #2

The manuscript "Isolation and Phenotypic Microarray Profiling of different Pseudomonas strains isolated from the rhizosphere of Curcuma longa L." by Parul Pathak, Ananya Naskar, Monika Singh, Sandeep Kumar Singh, Manoj Kumar Solanki, Nikunj Bhardwaj and Ajay Kumar deals with the isolation and characterisation of strains of the genus Pseudomonas which exhibited properties characteristic of plant growth promoting rhizobacteria.

Response: We are very thankful to the anonymous learned reviewer for their critical suggestions and comments. Definitely these suggestions improved the quality of our article. We have critically revised the article as per suggestions and the revised texts are highlighted in red.

Comments:

Line 57-58. References 5,9 are inconsistent with the statement made. I recommend deleting this sentence completely as it has no scientific basis.

Response: Deleted the sentence

The introduction is poorly structured. For example, the definition of PGPR is on lines 46-48 and lines 80-82.

Response: Thanks for suggestion, we have revised it

Line 88 is "PGPB" - correct the typo.

Response:Done

At the end of the introduction, there is no clearly stated purpose of the study.

Response: Thanks for suggestion, we have revised it

Materials and methods.

P.2.6 - missing reference to sequences (e.g. GenBank) used for analysis.

Response: Thanks for suggestion, we have added the citation

Line 154 - reference 6 - reference numbering appears to be incorrect.

Response: Thanks for remark, we have corrected this error

Figure 1 - photographs of colonies of microorganisms should not be shown. Furthermore, morphological characterisation of colonies is a very outdated method of characterising microorganisms.

Response: Thanks for remark. We totally agree with your remark, but the traditional methods are also a need to validate or start of the experiment. We will definitely consider your recommendation for our future study.

Line 197 - 16 S rRNA is written in a slit - probably a typo.

Response: Done

Table 2. The given sequences are short and small enough to determine the species of the strains on the basis of them. It is correct to compare your sequences with the 16S rRNA gene sequences of the type strains.

Response: Thanks for suggestion, we have revised and updated the table.

Figure 2 does not show the scale.

Response: Thanks for suggestion; we have replaced the fig.2 with new one

Figure 3 is not informative, I recommend deleting it.

Response: done

Figure 4 is of very poor quality and lacks transcripts A, B, It is not clear what the authors are trying to show with these data.

 Response: Thanks for the remark, however, through this figure we just want to show the similarity and dissimilarity of strains potential during antibiotic resistance

It should be noted that the Results and Discussion section is also not very well structured. I recommend rewriting this section by breaking the text into parts with justification of why certain methods are used.

Overall conclusion: I recommend a thorough revision of the paper. Authors should clearly state the purpose of the research. It is also important that the paper lacks scientific novelty. I recommend that the authors decide on the novelty and purpose and rewrite the text of the paper accordingly.

Response: Thanks for the comments. Now, we have thoroughly revised the article as per the comments of all the three anonymous learned reviewers. Hope this revised version improved the article slandered.

Reviewer 3 Report

Comments and Suggestions for Authors

Comments on the review manuscript stresses-2644819 entitled” Isolation and Phenotypic Microarray Profiling of different Pseudomonas strains isolated from the rhizosphere of Curcuma

longa L.”

                The paper presents the isolation and qualitative analysis of bacteria from the root zone of turmeric. The isolated bacterial strains were analysed for traits that promote plant growth (uptake of biogenic phosphorus and nitrogen) and increase resistance to salinity, for example. The authors also tested the resistance of the isolated strains to antibiotics. The results obtained are promising, but have not yet been tested in practice; their evaluation should be carried out on an experimental basis. For now, they are potential plant growth promoters. It is hoped that in future studies the authors will attempt field experiments to test the identified characteristics of the labelled strains.

                The results on the antibiotic resistance of the selected strains are interesting and may lead to the expression of these genes in other bacteria present in the soil.

Comments on the Quality of English Language

The work should be read carefully to correct minor spelling errors marked in the text.

Author Response

Reviewer #3

Comments on the review manuscript stresses-2644819 entitled” Isolation and Phenotypic Microarray Profiling of different Pseudomonas strains isolated from the rhizosphere of Curcuma longa L.”

                The paper presents the isolation and qualitative analysis of bacteria from the root zone of turmeric. The isolated bacterial strains were analysed for traits that promote plant growth (uptake of biogenic phosphorus and nitrogen) and increase resistance to salinity, for example. The authors also tested the resistance of the isolated strains to antibiotics. The results obtained are promising, but have not yet been tested in practice; their evaluation should be carried out on an experimental basis. For now, they are potential plant growth promoters. It is hoped that in future studies the authors will attempt field experiments to test the identified characteristics of the labelled strains.

                The results on the antibiotic resistance of the selected strains are interesting and may lead to the expression of these genes in other bacteria present in the soil.

Response: We are very thankful to the anonymous learned reviewer for their remark and suggestions, we have revised the article critically and definitely, we will evaluate the potential of strains in the field study.

Round 2

Reviewer 2 Report

Comments and Suggestions for Authors

The introduction is still poorly structured. The authors have aligned the aim with the comments. However, the formulation of the objective as presented in the article lacks novelty. 

Regarding the identification of strains - for example, a sequence of 712 bp is presented for strain (https://www.ncbi.nlm.nih.gov/nuccore/KM067138), but this is insufficient for species identification. Thus, the data presented on the taxonomic characterisation of isolates raise certain doubts. 

Comments on the Quality of English Language

 Moderate editing of English language required

Author Response

Reviewers#1

The introduction is still poorly structured. The authors have aligned the aim with the comments. However, the formulation of the objective as presented in the article lacks novelty. 

Response: We are very thankful to the anonymous learned reviewer for their critical suggestions and comments. Definitely these suggestions improved the quality of our article. We have critically revised the article as per suggestions and the revised texts are highlighted in red. Regarding your comments about the introduction and  formulation of objective, we have revised whole the introduction and objective  section in the revised article.

Regarding the identification of strains - for example, a sequence of 712 bp is presented for strain (https://www.ncbi.nlm.nih.gov/nuccore/KM067138), but this is insufficient for species identification. Thus, the data presented on the taxonomic characterisation of isolates raise certain doubts. 

Response, Thanks for the comments, However , we have already performed and followed Bergey’s Manual of Determinative Bacteriology  to identify the strains. Regarding your concern about the 712 bp. As the sequencer is not available in our institute, we have outsourced our samples for sequencing  and this can cause some times contamination/error. Because we have trimmed  some sequences due of low intensity and this cause reduction in bp from more than 1000 bp to 712.

Hope you understand our point or situation.

Round 3

Reviewer 2 Report

Comments and Suggestions for Authors

The authors have made some changes to the introduction. However, the taxonomic identification of the strains is incorrect. The lack of a sequencer is not an explanation for incorrect data. The paper lacks novelty as such. However, considering that the authors have done a good job in analysing the antibiotic resistance of the strains, it is possible to accept the article for publication in this form if the editor-in-chief of the journal deems it possible.

Comments on the Quality of English Language

Minor editing of English language required